# Dengue transmission dynamics in an urban setting in western India

**Karuppusamy Balasubramani[1☯], Syed Shah Areeb Hussain[2☯], Sushant Anil Sawant[3☯], Abhishek Govekar[4], Pooja Telugu Prakash[5], Aparna Naik[4], Debattam Mazumdar[4], Jagannath Nayak[4], Kuldeep Singh[2], Lokesh Kori[6], Kalpana Mahatme[7], Kumar Arun Prasad[1], Praveen Balabaskaran Nina[5]\*, Ajeet Kumar Mohanty[ID][4]\***

**1** Department of Geography, Central University of Tamil Nadu, Thiruvarur, Tamil Nadu, India, **2** ICMR-National Institute of Malaria Research, Dwarka, New Delhi, India, **3** School of Life Sciences, JSS Academy of Higher Education and Research, Mysore, Karnataka, India, **4** ICMR – National Institute of Malaria Research Field Unit, DHS Building, Campal, Panaji, Goa, India, **5** Department of Public Health and Community Medicine, Central University of Kerala, Kasaragod, Kerala, India, **6** Division of Communicable Diseases, Indian Council of Medical Research, New Delhi, India, **7** Directorate of Health Services Building, National Vector Borne Disease Control Programme, Campal, Panaji, Goa, India

☯ These authors contributed equally to this work.
\* ajeet.nimr@gmail.com (AKM); praveen.nina@cukerala.ac.in (PBN)

## Abstract

### Background

Goa state, located on India's western coast, has seen an increase in dengue cases in the last decade. Systematic characterization of case trends, serotypes, affected demographics, spatiotemporal clusters and hot spots, and environmental determinants was undertaken to guide evidence-based dengue prevention and control policies in Goa and across western India.

### Method

A health center level spatiotemporal analysis of dengue from 2011–2024 was performed using passive surveillance data routinely collected from all 34 health facilities through the National Vector Borne Disease Control Programme, Directorate of Health Services, Goa. The dengue trends were analyzed using the Seasonal Mann-Kendall (SMK) test. The space-time trends, dengue case clusters (high and low transmission zones across Goa), and forest-based forecasting were performed using the space-time pattern mining framework in ArcGIS Pro 3.x. The negative binomial generalized linear model with log link was used to quantify location effects. The correlation between climate change and rising dengue incidences was determined using a distributed lag non-linear model.

**Data availability statement:** The data that support the findings of this study were used from the National Centre for Vector-borne Disease Control, Directorate of Health Services, Goa (nvbdpgoa@gmail.com). Hence, the data access is restricted and will be available only upon request.

**Funding:** The author(s) received no specific funding for this work.

**Competing interests:** The authors have declared that no competing interests exist.

## Results

The dominant dengue virus serotype in Goa was DENV-2, accounting for 58.6% of infections, followed by DENV-1 (21.51%), DENV-3 (17.87%), and DENV-4 (2.03%). The clusters of dengue cases were predominantly observed in North Goa—the SMK test showed a significant positive trend across 16 of the 17 health facilities in the district. The space-time analysis showed a significant monotonic increase (standardized Mann-Kendall Z-statistics $\approx 3.94$; $p < 0.001$) of dengue cases in recurrent high-high clusters in Candolim, Porvorim, Siolim, Pernem, Panaji, Saligao, Mapusa, and Vasco health centers. Forest-based forecasting for 2025–2029 predicts consistent case-loads in the high-high cluster, with an average annual increase of ~21% expected cases. The regression model showed the significance of climatic variables with a lag period of 2–3 months and a total annual rainfall threshold of 630 mm in North Goa and 607 mm in South Goa. Rainfall above this threshold could lead to higher dengue transmission.

## Conclusion

Integrating space-time analytics, negative binomial modelling, and climate-lagged associations produced operationally useful risk maps and short-term forecasts. These outputs justify pre-monsoon source reduction, targeted vector control, serotype-guided surveillance, and climate-informed early warning for dengue in Goa and comparable settings in western India.

### Author summary

Goa is one of India's popular tourist destinations, attracting visitors from all over the world. The state also hosts a large number of migrant workers seeking employment in the expanding construction and hospitality sectors. This persistent flow of people, combined with Goa's tropical coastal climate that favors mosquito breeding, creates ideal conditions for the transmission of mosquito-borne diseases like dengue. Our study analyzed 14 years of passive surveillance data (2011–2024) to understand dengue dynamics across the state. We observed a high dengue burden in health centers across the urban coastal region, particularly in North Goa's Panjim, Mapusa, Candolim, and Porvorim. Our study, through short-term forecasting models, could identify high-risk transmission zones, thus aiding dengue control policies in Goa and similar coastal regions in western India.

## Introduction

Since its discovery in the late 1700s [12] dengue infection has emerged as a significant global public health concern, placing a severe socioeconomic burden on

healthcare systems, particularly in underdeveloped countries [3]. The disease has undergone a rapid global expansion from only 9 dengue-endemic countries before the 1970s to more than 100 by 2024 [4,5]. Over the past two and a half decades, there has been a steep 29-fold increase in dengue burden across the world, and in 2024, a staggering 14.6 million cases and 1200 deaths were recorded [5]. The rapid expansion and rising burden of dengue have underscored its status as a critical global health threat, necessitating urgent action and coordinated international efforts for control and prevention.

Dengue infection is hyperendemic in tropical and subtropical regions, with a majority of the cases reported from South Asia, Southeast Asia, and tropical Latin America [6]. In India, dengue was first reported from Calcutta in 1963 [7] and has since spread to many parts of the country, with most cases reported from Karnataka, Tamil Nadu, Kerala, Maharashtra, Uttar Pradesh, Rajasthan, Delhi, Madhya Pradesh, Bihar, and Telangana in 2024 [8]. Due to the considerable burden of asymptomatic cases in India [9], dengue is substantially under-reported, and the actual burden of dengue in the country is poorly quantified [10,11]. A nationwide survey in 2017–18 found 48.7% seropositivity for dengue in the Indian population [11]. Rapid urbanization, human and mosquito migration, rising insecticide resistance, and climate change are the key factors driving the spread of dengue across India and globally [12].

Goa, the coastal state in western India, is a popular tourist destination attracting millions of visitors every year. Goa's tropical climate is highly conducive to the breeding of *Aedes* mosquitoes and dengue transmission. Dengue was first reported in Goa in the early 2000s [13], and in the past two decades, its burden has increased steadily in the state, with 567 cases and 3 deaths reported in 2024 [8]. Dengue has been shown to impose a substantial economic burden on the affected regions, largely due to high costs of healthcare, loss of productivity, and reduced tourism revenue [14]. As the Goan economy is highly dependent on tourism, the establishment and spread of dengue in the state could have severe socio-economic consequences. In recent years, Goa has seen a significant expansion of dengue, and more and more health centers have started to report dengue cases. Nevertheless, Goa is still in the early stages of dengue endemicity; therefore, timely and targeted interventions can yet be highly effective in mitigating and potentially eliminating dengue risk. Developing a comprehensive understanding of the distribution of dengue incidence across Goa and identifying the facilitating environmental factors is critical for decision-making, resource allocation, and targeted intervention strategies. Given the background, the analysis maps the spatiotemporal distribution of reported dengue across Goa's health centers. Persistent hotspots and comparatively lower-risk zones are delineated. The contribution of environmental drivers to recent increases in transmission is quantified.

## Methodology

### Study site

Goa, India's smallest state (3702 sq. km), is situated along the western coastline of India, bound by latitudes 14°54'15" N to 15°47'30" N and longitudes 73°40'45" E to 74°20'27" E. Goa has a coastline that stretches 105 km along the Arabian Sea. The state's population is estimated to be 1.5 million, with 62% residing in urban areas. The state is administratively divided into North Goa and South Goa districts, comprising 13 talukas and 420 local administrative units. Goa receives an average annual rainfall of ~3300 mm, with ~90% of the rainfall occurring during the southwest monsoon season (June to September). The hottest month is May, when humidity is high, and temperatures exceed 35 °C (95 °F) during the day.

### Data source

The annual dengue case records from 2011 to 2024 were sourced from all the 34 health centers across Goa: 24 Primary Health Centers (PHCs), 04 Urban Health Centers (UHCs), and 06 Community Health Centers (CHCs) throughout the north and south Goa districts, monthly caseloads from individual health centers (2012–24), data by age and sex (2021–2024) along with dengue serotype data (2019–2024) were obtained from the National Vector Borne Disease Control

Programme (NVBDCP), Directorate of Health Services (DHS), Goa. No missing values were observed in the caseload datasets. The health center-wise estimated population (Census 2011) and total dengue cases reported during 2011–2024 are presented in S1 Table. Annual dengue incidence for 2011–2024 and average annual dengue incidence at each health center were calculated using the estimated population with 95% confidence interval.

Meteorological data were obtained from the ECMWF Reanalysis global dataset v5 (ERA-5), at 1 km resolution [15]. The hourly data of the temperature (K), rainfall (mm), relative humidity (%), wind speed (m/s), and atmospheric pressure (bar) from 2011 to 2024 were extracted in raster format, and monthly minimum, maximum, and averages for each of these variables were computed for the study period (only monthly total computed for rainfall). Temperature values were converted from Kelvin to Celsius for reporting the results.

The Landsat satellite images for the study area were obtained from the USGS Earth Explorer site (https://earthexplorer.usgs.gov/). The Landsat 5 images for February 1991 and the Landsat 9 images for February 2024 were used for the analysis of land use/land cover (LU/LC) change, while the Landsat 9 images of 2019–2024 were used to compute a built-up index for the study region (S2 and S3 Tables).

Health center locations and service areas were obtained from the District Health Office and used consistently for all years of analysis. No officially notified major changes in health center boundaries were reported during the study period; however, minor population variations in service catchment areas cannot be ruled out. Administrative boundary shapefiles used as base layers in all maps were obtained from the Survey of India (SoI), Government of India (https://onlinemaps.surveyofindia.gov.in/).

## Analysis of temporal trends of dengue in Goa

Annual dengue incidence and average annual incidence for Goa and the health centers were calculated using the estimated population from the Census 2011. To analyze the monotonous temporal trend in dengue incidence, a seasonal Mann-Kendall (SMK) test was applied to the monthly dengue case incidence data, and the significance of temporal trends at each of the 34 health centers was analyzed. The SMK test is a non-parametric statistical test that can detect monotonous trends in the data. The test doesn't make any assumptions regarding the underlying distribution of the data and is therefore well-suited for assessing temporal trends in dengue, which has recently emerged in Goa. In addition, the SMK test also adds an additional step to account for removing all seasonal effects by stratifying the data in identical seasonal blocks, thereby increasing its accuracy in identifying trends in seasonally transmitted infections like dengue. As significant autocorrelation in dengue cases was identified, the modified SMK test using the variance correction approach proposed by Hamed and Rao (1998) was applied, which detrends the data and adjusts the effective sample size using ranks of significant serial correlation coefficients [16]. The trends identified by the SMK test were considered significant at the 95% confidence interval ($p < 0.05$).

## Space-time analysis of dengue in Goa

The dengue data by health centers (2011–2024) were converted into a spatial database using the geographical coordinates values of each health center using the ArcGIS 10.8 software. The village boundaries under the health centers were traced from the 2011 Census Handbooks, and the location-wise data were transformed into spatial units (polygons) for space-time analysis. The geostatistical analysis was performed in ArcGIS Pro 3.x using the Space–Time Pattern Mining framework. The annual dengue counts (2011–2024) for the 34 health center jurisdictions in Goa were used to construct a space-time cube to perform local outlier analysis and time-series clustering. Local outlier analysis was attempted using Anselin Local Moran's I for each health center unit, and the units were classified into high–high, low–low, high–low, and low–high categories. Statistical significance was derived from 999 Monte Carlo permutations with Benjamini–Hochberg false-discovery-rate control ($α = 0.05$) within each year. We summarized each jurisdiction's multi-year behavior by frequency and persistence of cluster or outlier status. We applied agglomerative hierarchical clustering (Ward linkage) for time-series clustering after

z-standardization and estimation of correlation-based shape distance (1−Pearson's r). The number of clusters (k) was selected using silhouette width, within-cluster dispersion, and bootstrap stability (500 resamples; Jaccard index). Cluster-level temporal directionality was assessed with the Mann–Kendall test on the mean trajectory.

Time-series forecasting (five-year horizon) was conducted with the forest-based forecast tool using ensemble regression trees, training the health center-wise data (seed = 37545; 100 trees; sample size = 100%) from 2011 to 2023. The forest-based forecasting model used lagged dengue case counts (1–3 years) to capture temporal dependence, recent observed counts to represent short-term dynamics, a temporal index (year) to account for long-term trends, and health center identifiers to incorporate spatial heterogeneity across health centers. Despite the health center level time series containing a limited set of annual observations (13), the forest-based forecasting model employed in ArcGIS Pro is explicitly tailored to manage short, noisy, and non-linear temporal data. The ensemble structure reduces overfitting by aggregating predictions across multiple regression trees and leveraging shared temporal patterns across the health centers. The forest-based forecasting model incorporates multi-year lagged predictors (2–3 years) to capture long-term temporal dependence and inter-annual trends in dengue incidence at the health center level. Models were validated using 2024 data and used to forecast 2025–2029 with 95% prediction intervals. Accuracy was summarized by Root Mean Square Error (RMSE) for validation and by forecast uncertainty width (S4 Table). Automatic lag discovery (time-window range 2–3 years; mean 2.59; median 3) and seasonality detection were enabled using a "build by value" approach. Models produced moderate, spatially heterogeneous error (forecast RMSE 0.38–40.83, mean 7.79, median 4.68; validation RMSE 0.06–43.55, mean 11.93, median 6.69), indicating right-skewed difficulty concentrated in a few locations. The wide range of RMSE values across PHCs reflects heterogeneity in dengue incidence patterns, with lower RMSE observed in PHCs reporting consistently low case counts and higher RMSE in PHCs characterized by episodic outbreaks and greater inter-annual variability. Differences in data completeness and signal-to-noise ratios at the PHC level may further contribute to this variability. Automatic lags confirmed short-memory dynamics (2–3 years), and seasonality was identified in 41.18% of the sites. Feature importance was dominated by the first-order lag (~40% of total importance), consistent with strong one-year persistence. Outliers were flagged in 11 locations (32.35%), with the largest concentration during 2018–2019, contributing to the heavier upper tail of errors. To reduce overfitting, forecasting was performed using an ensemble of regression trees, which stabilizes predictions through aggregation. Model performance was assessed using temporal hold-out validation, with data from 2011–2023 used for model training and 2024 reserved for out-of-sample validation.

A negative binomial generalized linear model (log link) was performed in Python 3.x, and quantified temporal trend and between-health center differences were estimated and reported as coefficients and incidence rate ratios (IRRs) with 95% confidence intervals. All geoprocessing steps, scripts, and outputs were maintained under version control to ensure reproducibility. To quantify temporal trend and location effects while accommodating overdispersion, we fitted the following model with a log link.

$$\log\{E(Y_{it})\} = \beta_0 + \beta_1 \, \text{Year}_t + \sum_{j=1}^{J-1} \gamma_j \, \mathbb{I}(\text{PHC}_i = j),$$

where $Y_{it}$ is the annual count for health center $i$ in year $t$. Parameters were estimated by iteratively reweighted least squares. Results are shown as regression coefficients and incidence rate ratios (IRR = $e^\beta$), with 95% confidence intervals and Wald z-tests. Model adequacy was assessed using log-likelihood, AIC/BIC (LL-based), deviance, Pearson $\chi^2$, and Cragg–Uhler pseudo-$R^2$. The model showed adequate global performance (log-likelihood is −1376; deviance is 622; Pearson $\chi^2$ is 616 with 441 residual degrees of freedom; Cragg–Uhler pseudo-$R^2$ is 0.595), indicating that the covariates captured a substantial proportion of variation in dengue incidence (S5 Table). Model fit was assessed using pseudo-$R^2$ and likelihood-based diagnostics. Pseudo-$R^2$ values represent relative improvement over a null model and are not directly comparable to ordinary least squares $R^2$; higher values indicate stronger explanatory power for count data models.

Initially, a Poisson generalized linear model (GLM) was evaluated for modeling dengue case counts. However, diagnostic assessment indicated substantial overdispersion, with the variance exceeding the mean and the residual deviance-to-degrees-of-freedom ratio greater than one. Consequently, a negative binomial GLM was adopted to appropriately account for overdispersion in the count data. The negative binomial GLM assumes conditional independence of observations across health centers. In this study, the model was applied as a baseline regression framework to estimate covariate associations while accounting for overdispersion in dengue counts.

## Assessing the impact of climatic factors on dengue seasonality

Monthly reported dengue cases were plotted against monthly average temperature and monthly total rainfall, variables that play a major role in vector prevalence, to visually explore patterns that correspond to high dengue incidence and to provide insights into their roles in dengue transmission. Distributed lag non-linear models (DLNMs) were used to explore the potential non-linear and delayed associations between the 13 climatic predictors and dengue cases in the different health centers of Goa. The DLNM framework relies on a cross-basis matrix, a bi-dimensional function space that is the tensor product of two sets of basic functions, one of which describes the non-linear predictor/response relationship while the other describes the distributed lag structure of the relationship. The cross-basis matrix was constructed for each of the 13 climatic predictors using natural cubic splines with two knots placed at equally spaced quantiles over a period of six-month lag. The DLNM analysis focuses on short-term associations between rainfall and dengue incidence using monthly lag structures (0–6 months), allowing assessment of delayed effects operating over weeks to months.

Model parameters were estimated using the Bayesian spatiotemporal framework with integrated nested Laplace approximations (INLA). Spatial adjacency structure between the 34 health centers was constructed using the health center-wise boundary shapefile for Goa and fed into the INLA architecture. The health center-wise number of dengue cases from January 2012 to December 2024 was used as the response variable, assuming a negative binomial distribution. The cross-basis matrices of the 13 climatic predictors were used to specify the fixed effects. Random effects in the model were specified by a cyclic first order process for health center-wise monthly variations to capture seasonal patterns, as well as a Besag-York-Mollié (BYM2) spatial effect to account for the interannual variation at the PHC level. Natural logs of the population residing within each health center service area boundary were used as offsets for the model.

Initially the full model was fitted using all 13 climatic predictors and model fit statistics such as deviance information criterion and Watanabe-Akaike information criterion were estimated. The 95% credible intervals of the fixed effect coefficients for each lag component in the full model were then inspected to identify any significant cross-basis effects. These were compared to the results of the multi-collinearity test using Spearman's rank correlation and VIF test ($r > 0.8$ and/or VIF > 10 indicating strong collinearity) to identify the most influential climatic predictors for dengue transmission in Goa.

Subsequently, single-variable models were constructed for each of these most significant climatic predictors for a detailed analysis of the independent non-linear and lagged effect of these variables on dengue incidence. Cross-predictions were then used to produce estimates of the relative risk of dengue incidence by exposure level and lag from each of the single variable models. The exposure-lag-response surfaces of each of the influential climatic variables were visualized using contour plots, lag specific relative risk for dengue was visualized as line plots. Finally, a self-exciting threshold autoregressive model was applied to these predictors so as to identify the threshold value that significantly affects dengue transmission.

## Land use change detection analysis

The Landsat images (1991 and 2024) of Goa were processed through an unsupervised classification algorithm (K-means classifier). The output clusters were evaluated and assigned suitable labels based on the level-1 LU/LC classification scheme. The accuracy of the classified LU/LC classes was estimated with the help of high-resolution Google Earth

imagery and published Google maps. A total of 80 random points were used to evaluate the classification accuracy. The Kappa Statistic for the 1991 map is 88.57% (0.87) and 92.50% (0.91) for 2024.

The Normalized Difference Built-up Index (NDBI), used to identify built-up or urban areas, was estimated for Goa in Google Earth Engine (GEE). It is calculated based on the spectral reflectance properties of urban surfaces, which typically reflect more in the Short-Wave Infrared (SWIR) band than in the Near-Infrared (NIR) band. The positive NDBI values are likely built-up areas (concrete, asphalt, rooftops) while the negative values likely denote vegetation, water bodies, or bare soil. We estimated a median NDBI value using Landsat-9 satellite images from 2019 to 2024. All the positive pixels were extracted and mapped spatially to understand the built-up spread. The health center-wise dengue case distributions were overlaid on the land use change and NDBI maps for a better spatial visualization (association) of dengue cases in the high built environments.

## Results

### Distribution of dengue cases in different health centers across Goa

The spatial distribution of health center-wise dengue cases during 2011–2024 in Goa is presented in Fig 1. The cases are heterogeneously distributed, predominantly in urban areas in the western coastal regions of Goa. Between 2011–2024, 12 deaths due to dengue were recorded, and the health centers across Goa recorded a steady rise in dengue cases, peaking in 2019 (n = 726), and a higher caseload was reported in North Goa (n = 2673) (S1 Table). Candolim PHC, Mapusa UHC, Colvale PHC, Siolim PHC, Aldona PHC, Valpoi CHC, Sanquelim CHC, Porvorim PHC, Pernem CHC, Saligao PHC, and Panaji UHC in North Goa District and Vasco UHC, Margao UHC, Shiroda PHC, Curtorim PHC, Canacona CHC, and Cortalim PHC, in the South Goa District reported >100 dengue cases. Of the 34 health centers in Goa, these 17 accounted for 82.14% (3,875) of the total dengue cases (4,717) between 2011 and 2024. Of these, Vasco UHC had the highest case load (563 cases), followed by Candolim PHC (521 cases) and Mapusa UHC (437 cases). The least caseload was reported by the Sanguem PHC, with just 6 cases over the study period (S1 Table).

The age- and sex-wise distribution of dengue cases over the last four years (2021–2024) shows the majority of cases occurred in males (61.5%, 1335/2171) and adults ≥15 years of age (79.7%, 1731/2171) (S6 Table). The monthly distribution of total dengue cases in Goa from 2012 to 2024 (Fig 2A) shows a high caseload during the southwest (June-September) and northeast monsoon (October-December) seasons, peaking in October, just after the end of the southwest monsoon rains.

From 2019 to 2024, dengue virus serotype data for 495 and 193 samples from North and South Goa, respectively, showed 58.57% of the samples were DENV-2 serotype, followed by DENV-1 (21.51%), DENV-3 (17.87.%), and DENV-4 (2.03%). No mixed infections of DENV serotypes were detected. In 2024, DENV-1 serotype accounted for 41.37% of the samples (Fig 2B and S7 Table).

### Temporal trends of dengue cases across Goa

Over the study period, dengue incidence has increased in both magnitude and distribution. From 2011 to 2024, the cumulative incidence per 100,000 inhabitants was estimated to be 1.8 (1.2–2.6), 2.7 (1.9–3.7), 13.7 (11.9–15.8), 11.6 (9.9–13.5), 20.1 (17.9–22.5), 10.3 (8.7–12.1), 16.1 (14.1–18.3), 23 (20.7–25.7), 49.9 (46.3–53.6), 25.8 (23.3–28.5), 44.5 (41.2–48.1), 30.5 (27.7–33.4), 35.3 (32.3–38.5), and 39.1 (36–42.4), respectively. The modified SMK test showed that most health centers in North Goa had a significant positive trend for dengue cases at the 95% confidence interval (except Valpoi and Betki). Betki PHC showed a non-significant negative trend (-1.41, p value = 0.16) while Corlim PHC showed a significant negative trend (-2.10, p value = 0.04). Corlim reported fewer infections since 2014, and in Betki, no new cases were reported until 2023. In contrast, of the 17 health centers in South Goa, nine showed a negative trend in dengue incidence, none of which was significant. Non-significant negative trends in these health centers might be due to dengue outbreaks in some years,

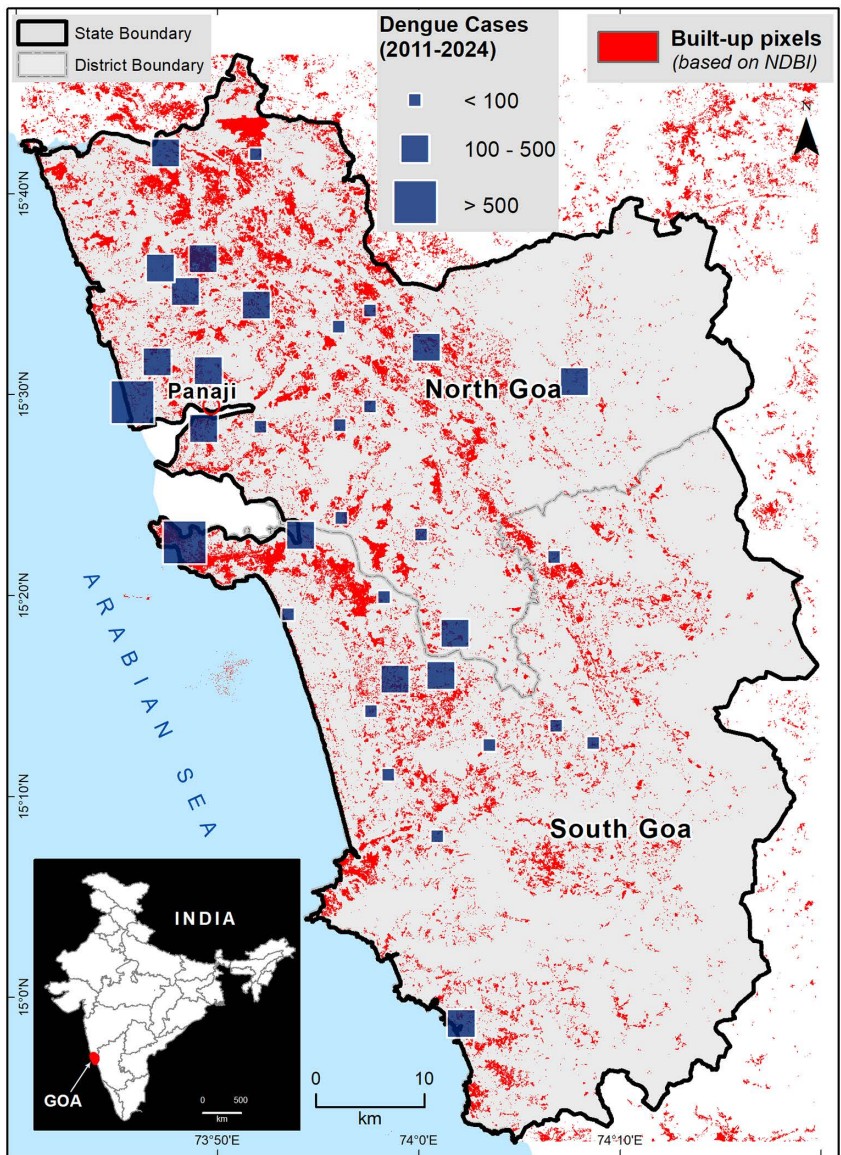

**Fig 1. Spatial distribution of health centre location-wise total dengue cases (2011-2024) in Goa with the positive Normalized Difference Built-up Index (NDBI) pixels representing the settlement distribution in the state.** The large (dark blue) squares denote high (> 500) caseload centers located within the dominant built-up (red) pixels. The inset figure represents the location of Goa state in India. The map is prepared using the administrative boundary shapefile, downloaded from the Survey of India (SoI), Government of India: https://onlinemaps.surveyofindia.gov.in/Product_Specification.aspx.

which tend to skew the SMK test results. A significant positive trend was found only in Cortalim (2.06, p-value = 0.04). In most of the North Goa health centers, a gradual rise in dengue incidence over the years has led to relatively stable trends. In contrast, South Goa is more prone to dengue outbreaks, resulting in non-significant trends (S8 Table).

## Space-time patterns of dengue in Goa

The Anselin Local Moran's *I* applied to annual case counts over 2011–2024 identified three behaviors across the health centers in Goa: (i) locations that alternated among cluster and outlier states (multiple types), (ii) locations that repeatedly

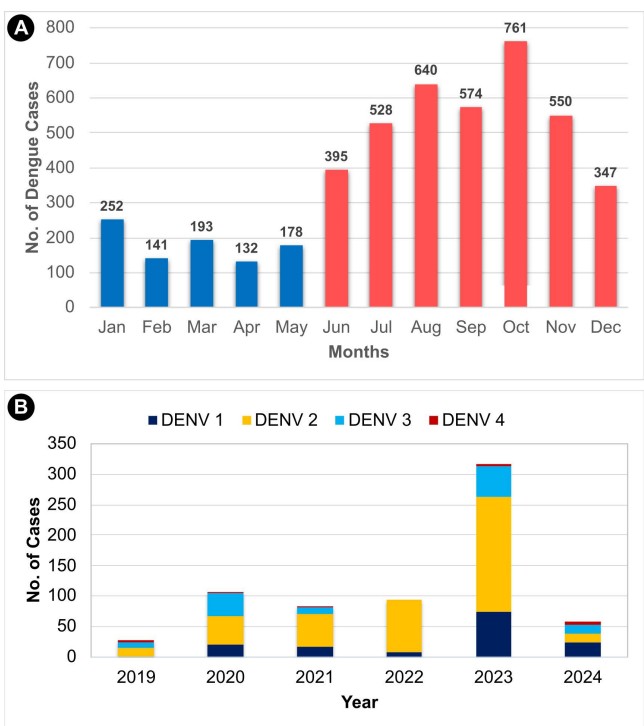

**Fig 2. A) Monthly distribution of total dengue cases in Goa from 2012 to 2024.** Blue and red color gradients represent lower caseloads during January–May and high caseloads during the rainy (monsoon) season of June–December, respectively. B) Prevalence of different serotypes of dengue in Goa from 2019 to 2024.

formed low–low (LL) clusters only, and (iii) locations that appeared exclusively as high–low (HL) outliers. No unit lacked spatial neighbors, and none registered an outlier in 2024. The spatial distribution of these clusters is shown in Fig 3A. The multiple-type clusters comprised 58.8% of health center units and accounted for all detected high–high (HH) cluster years as well as most outlier years [7 high–low (HL) and 10 low–high (LH)]. The recurrent HH clustering occurred most often in Candolim, Porvorim, and Siolim (four HH years each), followed by Pernem, Panaji, Saligao, Mapusa, and Vasco (three HH years each). The HL outliers were recorded once in Margao, Curtorim, Mapusa, Aldona, Corlim, Candolim, and Bali, whereas the LH outliers were observed in Marcaim (two years), Saligao (two), Chimbel (two), and once in Mayem, Cansarvanem, Siolim, and Pernem. The LL-only class included 13/34 units (38.2%), where LL years were common but typically few per site—for example, Sanguem registered four LL years, and Quepem, Navelim, Dharbandora, and Sanquelim recorded three each; this group spans much of the interior and southern belt (Cansaulim, Quepem, Chinchinim, Navelim, Loutolim, Sanguem, Canacona, Curchorem, Dharbandora, Ponda, Betki, Sanquelim, and Shiroda) (Fig 3A).

Fig 3A shows a clear coastal–interior contrast. The north-western coastal and peri-urban corridor—Candolim–Panaji–Vasco through Porvorim, Siolim, Mapusa, Aldona, Chimbel, Corlim, Cansarvanem, Bicholim, and Pernem—is dominated by multiple-type behavior, consistent with episodic synchrony (HH years), interspersed with outlier episodes. In contrast, most interior and southern health center villages persist as LL-only, forming broad contiguous zones of relatively subdued incidence. The single HL-only signature (Valpoi) stands out as an isolated high against low-incidence neighbors.

Time-series clustering based on the correlation profiles has grouped all the spatial units into two distinct temporal regimes: cluster-1 with a significant increasing trend and cluster-2 with no significant trend (Fig 3B). The Mann–Kendall test applied to the mean series of each group showed a significant monotonic increase for cluster-1 (test statistic $\approx 3.94$; $p < 0.001$) and no significant monotonic trend for cluster-2 (test statistic $\approx 0.33$; $p = 0.74$). Thus, one set of regions, mainly

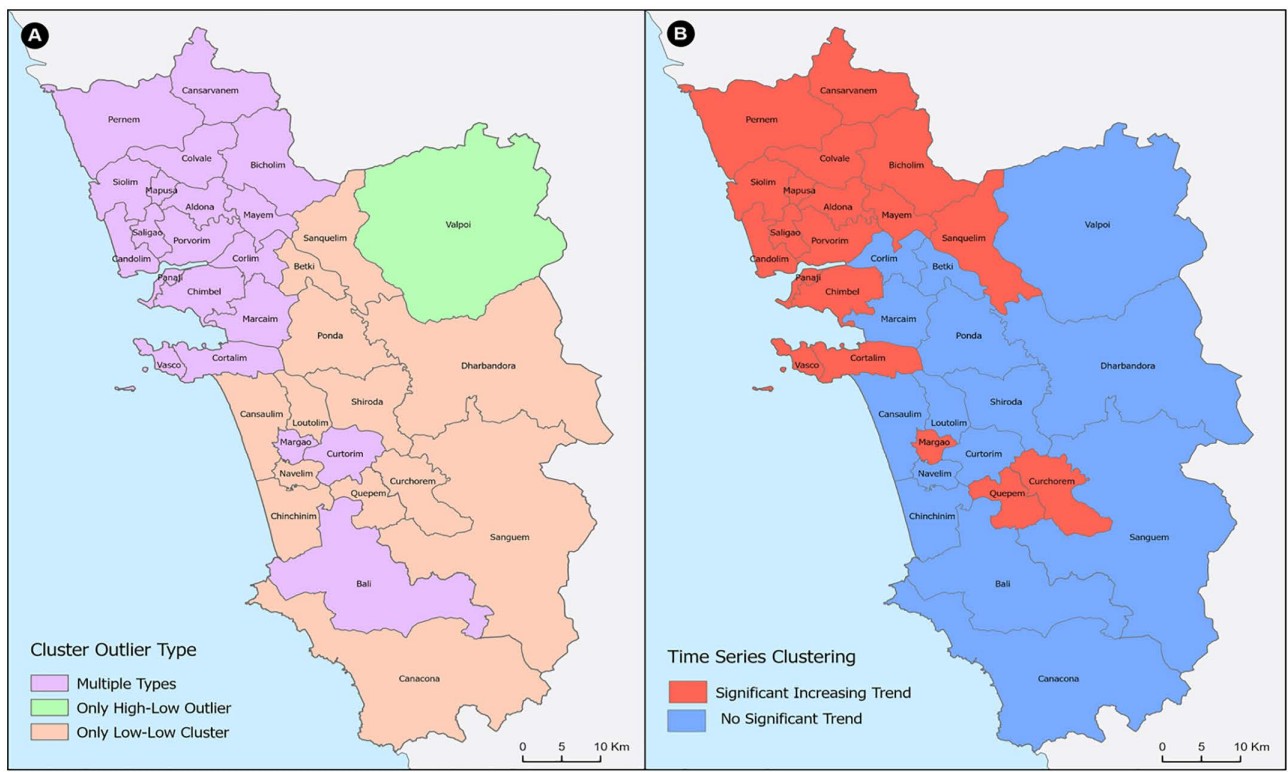

**Fig 3. Spatial distribution of A) cluster types and B) time-series trends in the clusters identified through the local outlier analysis of annual dengue incidence in Goa during 2011–2024.** The map is prepared using the administrative boundary shapefile, downloaded from the Survey of India (SoI), Government of India: https://onlinemaps.surveyofindia.gov.in/Digital_Product_Show.aspx.

the North Goa health centers, reported a coherent, multi-year case escalation pattern, whereas the South Goa cases (except a few health centers) fluctuate without a persistent directional change.

The mean and medoid trajectories (cluster representatives) corroborate these patterns: the increasing-trend cluster-1 rises in the latter half of the study window and remains elevated, while the no-trend cluster-2 exhibits lower-amplitude oscillations and episodic spikes that do not accumulate into a sustained rise (Fig 4). Taken together, the clustering reveals two temporally coherent dengue regimes—a coastal–peri-urban zone characterised by synchronised, multi-year intensification and an interior belt marked by fluctuation without directional drift—providing a concise, data-driven summary of how dengue dynamics have unfolded across Goa's health centers network from 2011 to 2024.

The Forest-based forecast of dengue cases for Goa in 2029 is shown in Fig 5. The case projection for 2029 preserved the historical patterns: higher projected burdens (> 40 cases) along the coastal/peri-urban corridor—notably in the vicinity of Siolim, Vasco, Cortalim, and Canacona centers. The adjacent regions of these centers fall in 20–40 cases, and the most southern centers remain < 20 cases except Canacona CHC which reported a caseload of >40. The health center-wise projection of dengue cases, along with the overall projection trajectory for Goa, is presented in Fig 6. The centers of consecutive dengue hotspots in Goa are projected to maintain consistent caseloads in the next five years (2025–2029). The projections appear to be governed by short-lag autoregression with occasional but influential spikes, yielding spatially coherent projections that prioritize intensified surveillance and vector control, especially along the coast and peri-urban areas.

The negative binomial generalized linear model fitted with a log link model showed that annual incidence data is a strong positive predictor of dengue incidence ($\beta = 0.1907$, $p < 0.001$). On the multiplicative scale, this corresponds to an

PLOS Neglected Tropical Diseases

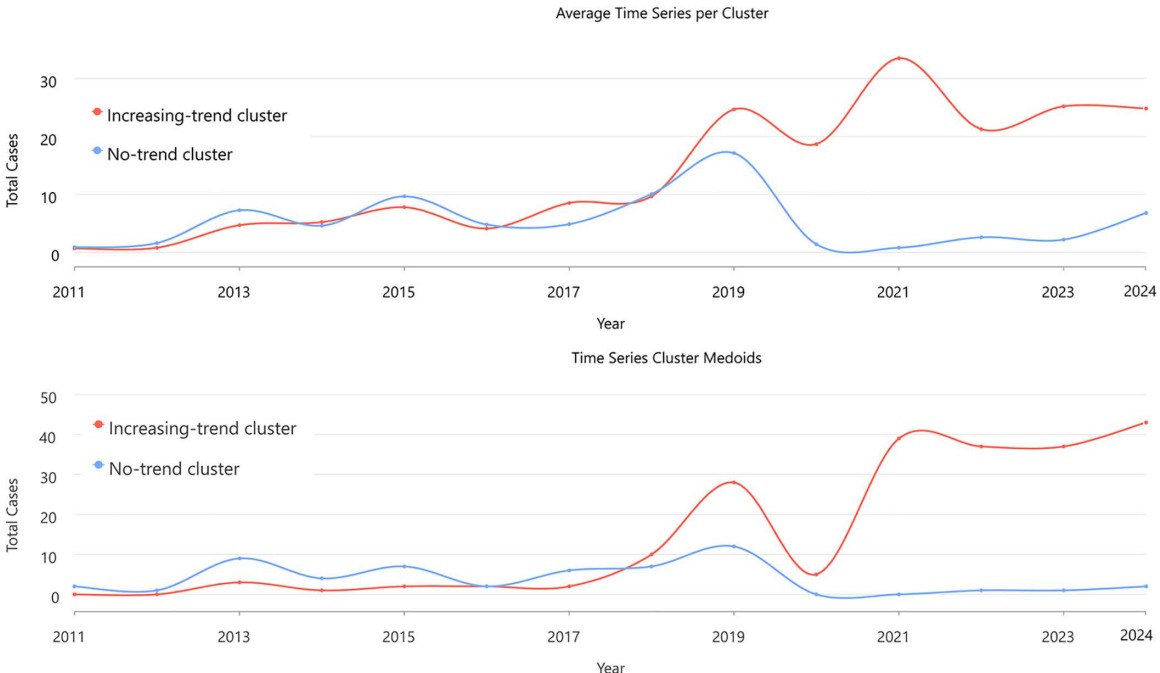

**Fig 4. Mean (top) and medoid (bottom) trajectories of dengue cases in the cluster representatives (increasing trend cluster and no-trend cluster) identified by the Mann–Kendall test using annual dengue incidence data of health centers in Goa during 2011–2024.**

incidence-rate ratio (IRR) of approximately 1.21 per year, or a 21% average annual increase in expected cases, holding location constant. The smoothed mean trend from the model rose steadily across the study period, whereas observed means displayed episodic peaks, consistent with superimposed outbreak activity. This pattern likely reflects unmeasured seasonal–environmental drivers and programmatic changes that vary between years (S5 Table).

Location effects revealed pronounced differences between the health centers after adjustment for time. Two coastal/peri-urban jurisdictions exhibited persistently elevated risk: Vasco ($\beta = 1.3277$, $p = 0.001$; IRR $\approx 3.77$) and Candolim ($\beta = 1.2160$, $p = 0.002$; IRR $\approx 3.38$). In contrast, several southern centers were associated with markedly lower incidence relative to the reference. The spatial gradient—higher risk in dense, tourism-linked coastal areas and lower risk in several interior jurisdictions—likely indicates differences in human mobility, housing density, and environmental suitability for *Aedes* breeding (S5 Table).

### Impact of climatic and land use factors on dengue transmission in Goa

A visual comparison of the seasonal patterns in dengue cases as well as rainfall in Goa is presented in Fig 7. During the northeast (post) monsoon period (October-December), the frequency of dengue cases nearly doubled, indicating the lag-effect (1–2 months) of rainfall on dengue cases in Goa. The high incidence of dengue is followed by high rainfall. In contrast, the monthly average temperature remains ambient throughout the year in both North and South Goa (between 20 °C and 30 °C), which has relatively little impact on the seasonality of dengue transmission.

Correlation analysis of dengue cases with climatic variables in each district did not yield any significantly high correlation, with all correlation coefficients <0.25. Multicollinearity test of the climatic predictors identified very strong significant collinearity between the maximum, minimum and average of the relative humidity (r = 0.71-0.88) as well as of the atmospheric pressure (r = 0.985-996) with each other (S1 and S2 Figs). The results from the full model showed that at least one

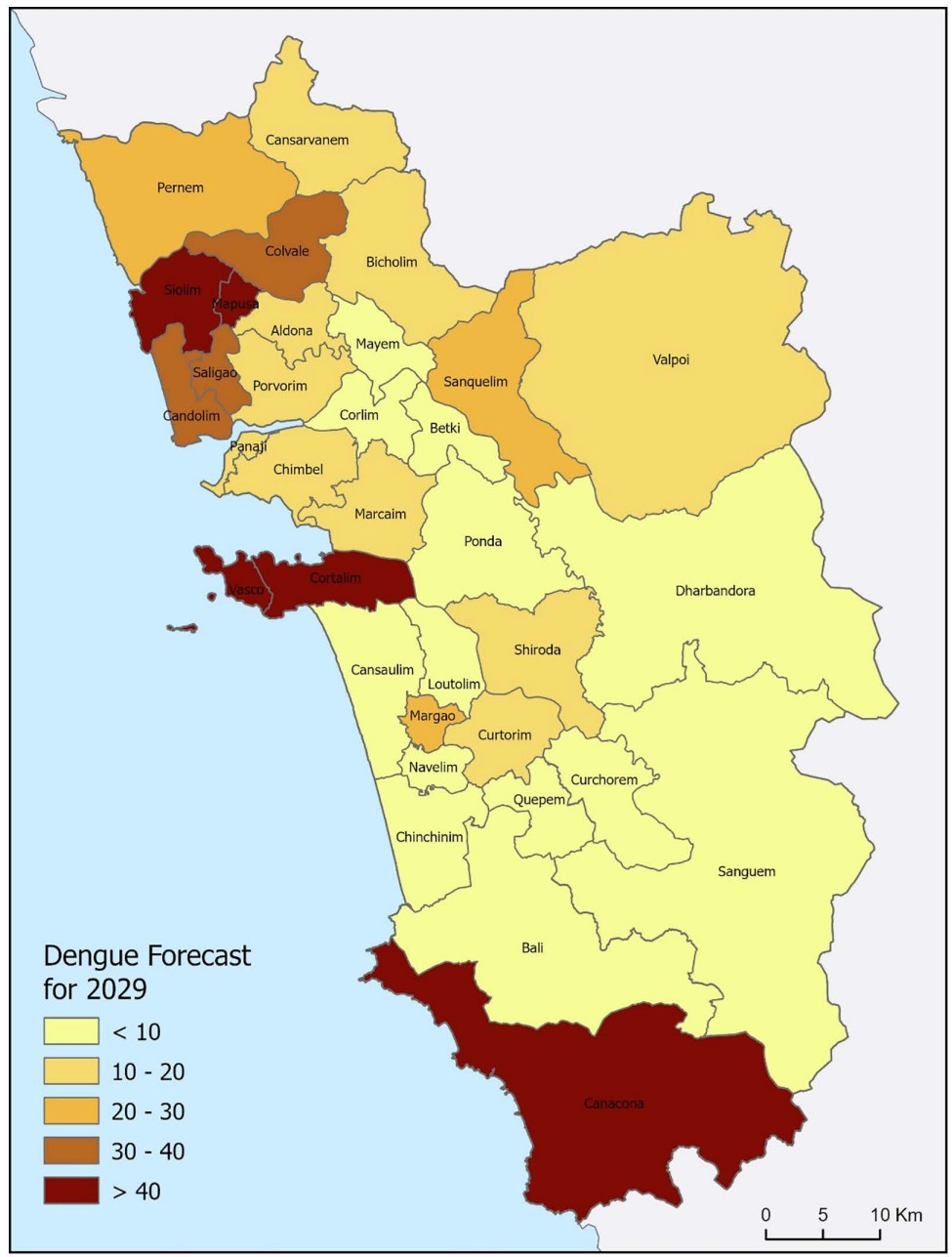

**Fig 5. Forest-based forecast of dengue cases in the health centers of Goa in 2029.** The darker shade denotes higher dengue caseload areas. The forecast uncertainty is quantified using 95% prediction intervals reported in the forecasting results. The map is prepared using the administrative boundary shapefile, downloaded from the Survey of India (SoI), Government of India: https://onlinemaps.surveyofindia.gov.in/Digital_Product_Show.aspx.

lag from each of the 13 climatic predictors had some significant effect on the model; however, the effect of minimum and maximum wind speed was negligible as compared to the other variables (< 1). This indicated that these variables did not play a significant role in dengue transmission. Moreover, among the atmospheric pressure and relative humidity variables, the maximum atmospheric pressure and minimum relative humidity had the highest mean absolute effect, hence these

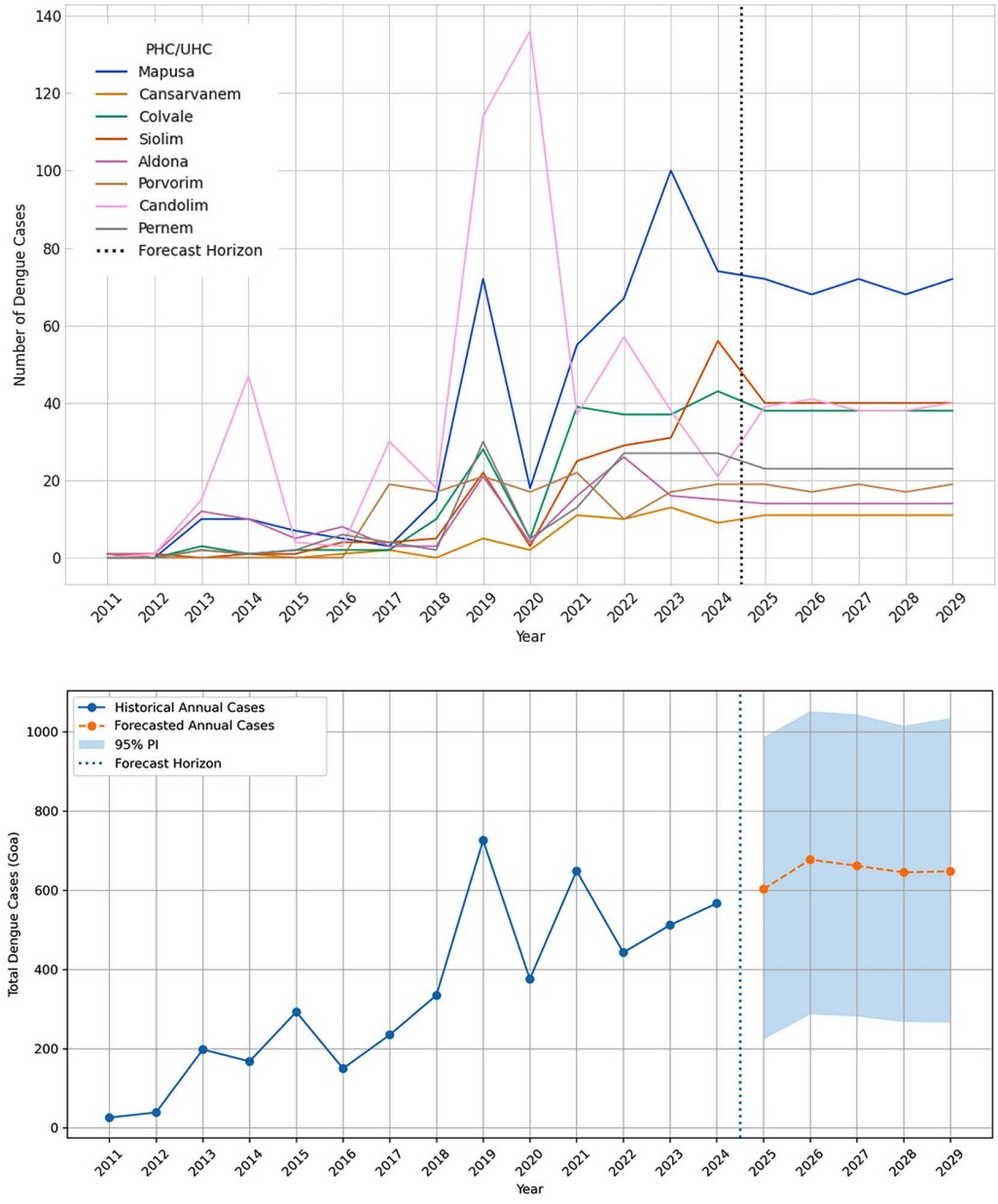

**Fig 6. A) Dengue forecast for the consecutive hotspot health centers (top) and B) overall dengue forecast (bottom) in Goa until 2029.**

were retained for the single variable models in addition to the minimum, maximum, and average temperature, total rainfall and average wind speed (S9 Table).

Among the single variable models, the total rainfall model had the lowest Deviation Information Criterion (DIC) value (11024) indicating that it had a very significant effect on dengue transmission (S10 Table). This was followed by the minimum relative humidity (13134). The other five climatic predictors had similar DIC values (~13175). Contour plots of the exposure-lag-response relationship of temperature variables with dengue incidence (S3 Fig) revealed that very high relative risk of dengue was only observed at 5–6 month lag for the maximum temperature and average temperature, while

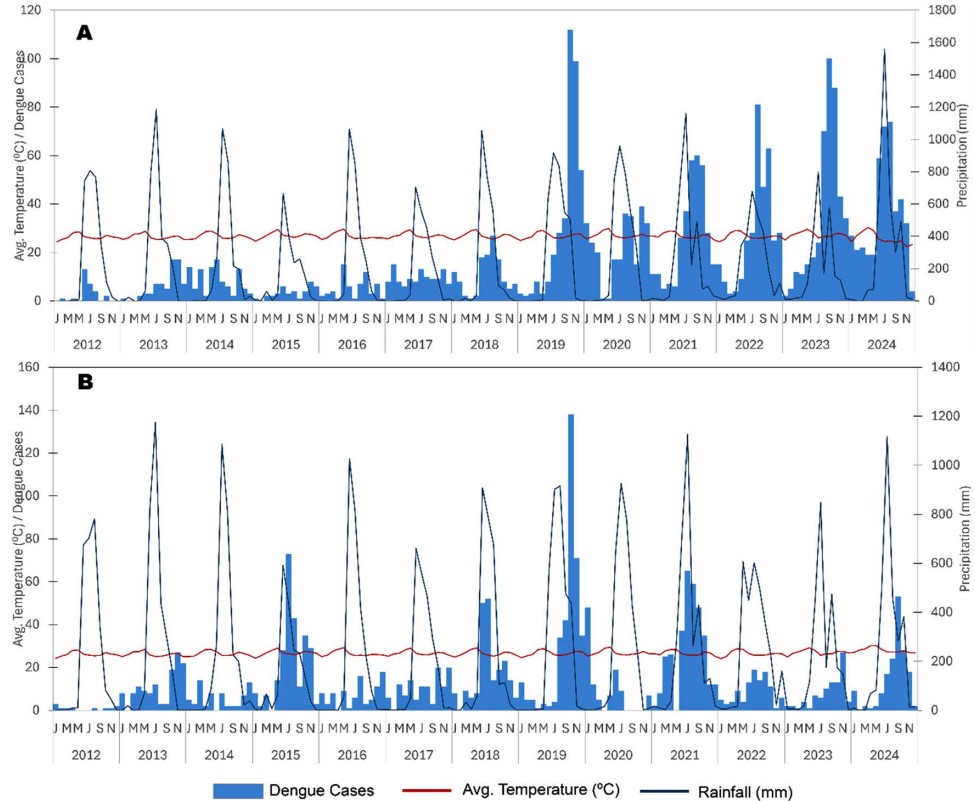

**Fig 7. Yearly dengue cases (2012-2024) trend correlation with climatic factors (temperature, rainfall) in A) North Goa and B) South Goa.**

no lag was most significant for the minimum temperature. The highest relative risk of dengue was observed between the range 23 – 27 $^0$C, with temperatures below 21 $^0$C being highly inimical to dengue risk. This shows that drops in minimum temperature may have sudden impacts on dengue transmission in Goa.

The total rainfall showed the most widespread impact on the relative risk for dengue at values above 800 mm for all lag periods, though the overall risk was highest at 3–4 months lags. The threshold for total rainfall that determines dengue transmission was found to be 630 mm in North Goa and 607 mm in South Goa. Rainfall greater than this threshold was found to result in strong autoregressive patterns, suggesting higher dengue transmission, whereas rainfall below this threshold resulted in significantly weaker autoregressive patterns. The identified rainfall thresholds (630 mm for North Goa and 607 mm for South Goa) correspond to monthly cumulative rainfall totals, consistent with the monthly temporal resolution of the DLNM analysis.

Extreme relative humidity was also associated with a higher relative risk for dengue at 2 months lag, while slightly lower relative humidity was associated with a moderate relative risk between 4–6 months lag. Lower maximum atmospheric pressure was associated with dengue risk between 3–6 month lag periods, while higher wind speeds (between 5–7 m/s) were associated with high dengue risk at 1–2 month lag period.

In the LU/LC comparison between 1991 and 2024, forest was the dominant class in both years, with 68% and 67% of total land area, respectively (Fig 8). Most of the agricultural land and forest area, particularly in the coastal parts, have been converted to built-up areas. There is a fourfold increase in build-up area from 1991 (~70 sq km and ~2% of total area) to 2024 (~260 sq. km and ~7% of total area). In addition to built-up area, pits and barren lands (labelled as misc in

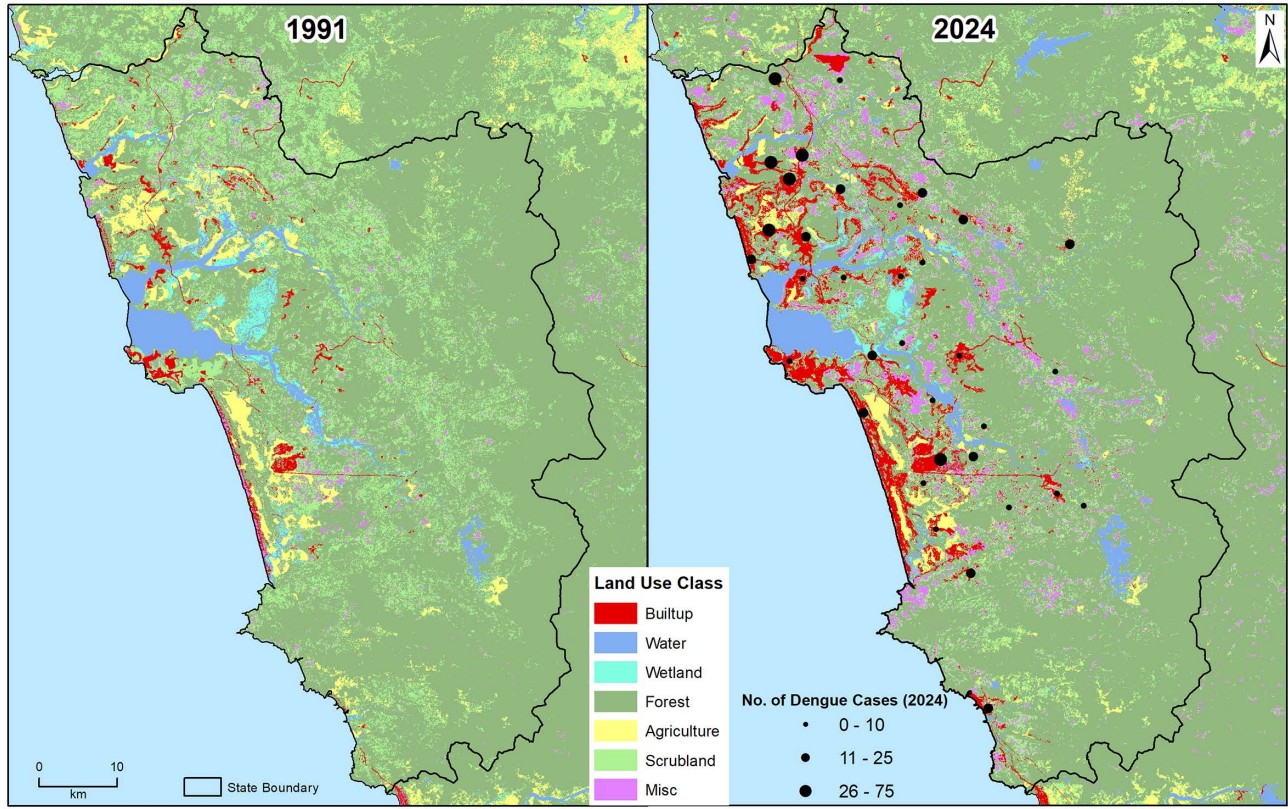

**Fig 8. Comparison of Goa's LU/LC (1991 vs. 2024).** The changes in LU/LC classes (in %) from a) 1991 (left) to b) 2024 (right) are overlaid on center bottom of the maps, denoting the color codes of the land use class. Forest (dark green) is the dominant LU/LC class (% in 1991 and % in 2024). The built-up (red) and miscellaneous (pink) classes show substantial changes from 1991 to 2024, both spatially and temporally. The map is prepared using the administrative boundary shapefile, downloaded from the Survey of India (SoI), Government of India: https://onlinemaps.surveyofindia.gov.in/Digital_Product_Show.aspx.

Fig 8 have also increased by replacing the scrub lands. The health centers that reported high dengue cases were situated in the highly built-up areas, and their surroundings, especially in coastal regions.

## Discussion

Since the early 2000s, dengue cases in Goa have been steadily increasing. The 4,717 cases reported during 2011–2024 highlight the emerging public health concern. Temporal trends in dengue incidence show that cases are on the rise across most health centers in North Goa, many of which have only recently emerged as epicenters for dengue transmission. The strong temporal clustering of dengue cases from 2021 through 2024 corroborates the upward trend in dengue incidence since 2020. The trend observed in Goa is in line with the increasing incidence of dengue in India; over the past decade, India has witnessed an 11-fold increase in dengue burden [17].

In this study, individuals aged 15 years and above, and males had a higher risk of dengue infection, a demographic skewness observed across the country [18–22]. In Goa, all four dengue virus serotypes were detected, with DENV-2 being predominant (58.57%). In addition to DENV-1 and DENV-2, Goa is also witnessing an increase in DENV-3 serotype. Recent studies from other south Indian states, including Karnataka and Kerala, have reported similar serotype shifts and emergence of DENV-3 dominance [23,24]. Studies from other parts of India have reported DENV-1 and DENV-2 as the

predominant serotypes [25,26], consistent with national data reported by Jagtap et al. (2023), which highlights DENV-2 as the dominant serotype across India since 2013 [27]. In Goa, the minor serotype is DENV-4, with a prevalence of 2.03%. DENV-4 has recently emerged in parts of South India; its low prevalence in our study may indicate regional variation or early-stage circulation in Goa. In several states, dengue infections are associated with more than one DENV serotype [25,28]. The absence of mixed infections in Goa may reflect localized transmission dynamics specific to our study region. Additionally, the limited sample size used for dengue serotyping may have hindered the detection of mixed infections. Further clinical studies would provide a more comprehensive understanding of the dengue serotypes circulating in Goa.

The co-circulation of all dengue serotypes in Goa poses a high risk of sequential heterotypic infections. The primary infections in dengue are mostly mild or asymptomatic and confer immunity to homologous serotypes [29]. In the subsequent infection with a different strain, pre-existing antibodies cross-react with the new strain but fail to neutralize it, leading to antibody-dependent enhancement (ADE), which increases viral load and disease severity [30]; ~90% of severe dengue cases reported are due to secondary/subsequent dengue infection from heterologous strains [31].

These immunological risks are intensified by the other social and ecological factors shaping dengue transmission in Goa. Our space-time analysis identified four major dengue outbreaks in Goa during the study period, along with several minor spikes in cases. Three out of four large outbreaks occurred along the coastline of North Goa, where highly industrialized and urbanized built-up areas have replaced forest and agricultural lands. The active construction sites in these areas generate abundant artificial water-holding sites that could serve as vector breeding sites and intensify human–vector contact in work camps and high-density housing clusters, facilitating the increased transmission of dengue. For instance, the four urban health centers in the state—Mapusa, Panaji, Vasco, and Margao continuously report high case-loads and account for 30.48% of all the dengue cases reported since 2011. These urban hotspots are driving dengue into rural parts of the state, as seen in the increasing caseloads at rural health centers in Canacona and Valpoi. Local Moran's *I* results also delineate a dual structure of dengue dynamics in Goa: 1) coastal–peri-urban corridor with recurrent HH clustering and occasional outliers, implying synchronized surges across neighboring units; and 2) extensive interior/southern tracts where incidence remains consistently low relative to similarly low neighbors. The HL-only signature in Valpoi suggests a localized driver or introduction pathway that does not generalize to adjacent areas. These statistically derived patterns align with the geographic distribution of major coastal tourist centers, which may contribute to the spatial synchrony observed there; however, corroboration would require explicit covariate analysis.

The cluster exhibiting the increasing trajectory is concentrated along the north-western coastal and peri-urban corridor, with contiguous membership spanning the Candolim–Panaji–Vasco axis and adjacent hinterlands. These regions are also predicted to have higher caseloads in 2029. Several pockets of the same regime appear in the south-western belt (e.g., around Margao, Cortalim, Quepem, and Curchorem), indicating localized synchrony with the coastal pattern. In contrast, the interior and eastern–southern tracts largely belong to the non-trend cluster, including broad areas centered on Valpoi, Dharbandora, Sanguem, and Canacona, where annual counts oscillate around comparatively stable baselines. The mapped overlay aligns spatially with the increasing-trend cluster along the coast, suggesting that mobility and population density may co-vary with the observed temporal dynamics, although causal attribution is beyond the scope of this classification exercise.

The primary dengue vector, *Aedes aegypti*, is widespread across Goa [13,32,33]. Its ability to breed in small man-made containers, resistance to overcrowding, and tolerance to large temperature ranges make it highly adapted to breeding in urban households [34] and have enabled it to out-compete *An. stephensi* in many settings [35,36]. Even though the bionomics of *Aedes* is different than *Anopheles*, another dominant vector species in Goa [32], both utilize urban water-storage infrastructure for breeding, highlighting the importance of poorly maintained man-made aquatic sites that could harbor multiple vector species [37,38]. Health centers reporting high dengue burden are located in coastal cities and towns that have witnessed a four-fold increase in built-up area since 1991, a result of rapid urbanization. In addition to the steep increase in the built-up areas, Goa also has numerous pits and barren lands that act as water reservoirs. These activities

have created a persistent ecological receptivity for breeding of *Aedes*, sustaining dengue transmission in urban and peri-urban regions of Goa.

The continuous expansion of the construction and hospitality sectors to sustain tourism has attracted scores of migrant workers from different parts of India to Goa. The malaria outbreaks in Goa are largely attributed to these migrant workers, who carry the *Plasmodium* parasites from malaria-endemic states [39]. It is plausible that these migrant workers from across India could have facilitated the spread of different dengue serotypes across Goa. A substantial proportion of dengue infections in India are asymptomatic, contributing to a large hidden pool of transmission. A community-based study showed asymptomatic dengue cases outnumbered clinical cases, and serological surveys revealed that subclinical infections comprised 40% of all dengue cases in Delhi [9,40]. According to the European Centre for Disease Prevention and Control, 40–80% of all dengue infections are asymptomatic, acting as a reservoir for silent transmission [41]. This is compounded by severe underreporting in India; only 0.35% of clinically diagnosed dengue cases are captured in national surveillance, and the actual burden is extremely higher than official figures [42]. The influx of migrant workers from hyperendemic regions with asymptomatic and viremic strains, coupled with ongoing construction-driven proliferation of *Aedes aegypti* in urban centers of Goa, could have facilitated dengue transmission (symptomatic and asymptomatic) to the naive local population. Hidden asymptomatic infections complicate disease surveillance and may increase the likelihood of severe infections when re-infected with heterologous serotypes. Therefore, integrated serological and molecular surveillance systems, extending beyond febrile cases, are crucial for detecting and mitigating these silent transmission chains, especially in urban clusters reporting dengue outbreaks.

The seasonality of dengue in Goa highlights the strong influence of rainfall, with case frequency peaking 1–2 months after the monsoon. The months where total annual rainfall exceeds the thresholds (630 mm in North Goa and 607 mm in South Goa) show a steep rise in dengue transmission, reflecting the role of rainfall-driven vector proliferation. The effect of temperature on dengue incidence was found to be minimal. The annual temperature in Goa ranges from 20 °C to 30 °C and does not exceed the tolerance limit of the *Aedes* vectors, which thrive between 25–30°C [43]. Even though *Aedes* vectors have been shown to be temperature-tolerant [44,45], the *Aedes* population in Goa does not have to undergo temperature-dependent physiological adaptations. Wind speed and relative humidity also play a modulatory role in Goa, with higher average wind speeds increasing transmission in the short term due to vector dispersal (1–3 month lag), but suppressing transmission in the long term (5–6 month lag), while high humidity at short lags appears to support vector activity [46].

The study uses three separate analytical methods—forest-based forecasting, climate-driven DLNM analysis, and hotspot detection to study different elements of dengue transmission. The different high-risk areas identified through various methods resulted from variations in data inputs, temporal scales, and model assumptions, as well as unobserved confounding factors. The present research did not implement a complete multi-model framework that would merge all output data into a single analytical framework, but future studies should develop this method to improve dengue risk evaluation.

Besides, the study has data limitations that need consideration. First, the data was analyzed for 14 years (2011–2024) due to the unavailability of earlier surveillance data. While this timeframe does not comprehensively capture long-term trends, dengue cases prior to 2010 in Goa were minimal and sporadic, which may not have significantly affected the outcome. Additionally, the analysis was conducted using monthly aggregated data rather than weekly observations, which may have limited the ability to detect short-term fluctuations. Nevertheless, the low number of dengue cases meant this effect may not be too significant. Second, the data rely only on passive surveillance, which is prone to substantial under-reporting, and many individuals with mild or asymptomatic conditions do not seek medical attention. The gross underreporting of dengue cases in India has also been highlighted in earlier studies [10]. This could lead to an underestimation of the absolute dengue burden in the state and may affect the precision of risk estimates and forecasting outputs, particularly in areas with variable healthcare access or diagnostic practices. However, the current analysis was based on routine surveillance data that are consistent throughout the study period; therefore, the spatial patterns and relative comparisons across locations and time remain informative. Third, the data utilized in our study were at the health center-level

and did not have details on patient location, making it unclear whether cases were locally acquired or imported from other dengue-endemic regions. The analysis assumes that the health center service areas have not changed throughout time. Although official records show no major changes in the distribution of health centers in Goa, unreported modifications in the service areas could affect detailed geographic patterns and should be considered when interpreting local cluster results. In addition, the socio-demographic profile, economic conditions, and vector distribution, which could influence dengue transmission, were not available in the obtained dataset. The serotypes were tested on a limited sample, which may not accurately reflect the actual distribution of serotypes in the state, and the presence of mixed serotype infections couldn't be captured.

Despite these limitations, the study presents important insights into the spatiotemporal distribution of dengue in Goa and the underlying environmental factors. These results will enable state vector-borne disease control authorities to strengthen the surveillance system by targeting the identified urban hotspots that facilitate the spatial expansion of dengue into rural regions. At the same time, key insights into the lagged effect of climatic variables provide a clear understanding of seasonality in dengue transmission, which will be critical for preparedness, active case monitoring, and effective application of interventions.

## Conclusion

This study highlights the spatiotemporal heterogeneity of dengue transmission in urban and industrialized areas of Goa, underscoring the role of land use changes in mosquito breeding and construction related migration in shaping disease transmission dynamics. In addition, it emphasizes the importance of integrating climatic surveillance, particularly rainfall, into early warning systems for dengue prevention. Strengthening both passive surveillance and active case detection, along with screening migrants from endemic areas, will be crucial for improving the accuracy of disease burden, which can aid in controlling disease transmission at the health center level. Screening migrant workers for dengue can reduce the importation of dengue cases from other endemic states. Building climate-informed models can guide timely vector control, optimize resource allocation, and strengthen community preparedness strategies. This approach could serve as a template for other coastal and monsoon-affected states in India, where similar climatic conditions favor dengue transmission. Future research should focus on integrating spatiotemporal climatic models with real-time case data and vector indices, using machine learning to build operational early warning tools.

## Supporting information

**S1 Fig. Multicollinearity between 13 climatic variables based on Pearson's correlation factor.**
(DOCX)

**S2 Fig. Scatter plots of the 13 climatic predictors against dengue incidence in all 34 PHC/UHCs in Goa to assess the patterns of relationship between the predictors and dengue incidence.**
(DOCX)

**S3 Fig. (A) Contour plots to show the exposure-lag-response of the single variable models for each of the seven most significant climatic predictors for dengue and (B) Lag dependent relative risk curve of dengue for each of these variables at the 10th, 50th and 90th percentiles.**
(DOCX)

**S1 Table. Dengue incidence data from 2011 to 2024 in the state of Goa.**
(DOCX)

**S2 Table. Metadata of satellite datasets used in the study.**
(DOCX)

**S3 Table. Land use/ land cover (LU/LC) area composition between 1991 and 2024.**
(DOCX)

**S4 Table. Summary of accuracy across locations and of time series outliers.**
(DOCX)

**S5 Table. Generalized linear model regression results.**
(DOCX)

**S6 Table. Age and sex wise distribution of dengue cases in North and South Goa districts.**
(DOCX)

**S7 Table. Distribution of dengue Serotypes in Goa during 2019–2024.**
(DOCX)

**S8 Table. Seasonal Mann-Kendall test for monotonous trends in 34 health centers of Goa.**
(DOCX)

**S9 Table. Correlation analysis of different meteorological variables with dengue cases in each of the 34 health centers in Goa.**
(DOCX)

**S10 Table. Deviation Information Criterion (DIC) and the performance log score of the single variable models.**
(DOCX)

## Acknowledgments

Authors are thankful to the Director, ICMR-National Institute of Malaria Research, New Delhi and Indian Council of Medical Research for institutional and infrastructure support. Authors are also thankful to the NVBDCP, Directorate of Health services, Goa for providing the monthly dengue data.

## Author contributions

**Conceptualization:** Praveen Balabaskaran Nina, Ajeet Kumar Mohanty.

**Data curation:** Abhishek Govekar, Pooja Telugu Prakash, Aparna Naik, Debattam Mazumdar, Jagannath Nayak, Praveen Balabaskaran Nina, Ajeet Kumar Mohanty.

**Formal analysis:** Karuppusamy Balasubramani, Syed Shah Areeb Hussain, Sushant Anil Sawant.

**Resources:** Praveen Balabaskaran Nina, Ajeet Kumar Mohanty.

**Software:** Karuppusamy Balasubramani, Syed Shah Areeb Hussain, Sushant Anil Sawant.

**Supervision:** Praveen Balabaskaran Nina, Ajeet Kumar Mohanty.

**Validation:** Karuppusamy Balasubramani.

**Visualization:** Praveen Balabaskaran Nina, Ajeet Kumar Mohanty.

**Writing – original draft:** Karuppusamy Balasubramani, Syed Shah Areeb Hussain, Sushant Anil Sawant, Praveen Balabaskaran Nina, Ajeet Kumar Mohanty.

**Writing – review & editing:** Abhishek Govekar, Pooja Telugu Prakash, Aparna Naik, Debattam Mazumdar, Jagannath Nayak, Kuldeep Singh, Lokesh Kori, Kalpana Mahatme, Kumar Arun Prasad.

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
