## [Decision Letter · Decision Letter 0]

26 Nov 2025

Dengue transmission dynamics in an urban setting in western India

Dear Dr. Mohanty,

Thank you for submitting your manuscript to PLOS Neglected Tropical Diseases. After careful consideration, we feel that it has merit but does not fully meet PLOS Neglected Tropical Diseases' publication criteria as it currently stands. Therefore, we invite you to submit a revised version of the manuscript that addresses the points raised during the review process.

We look forward to receiving your revised manuscript.

Kind regards,

Elda Eliza Sanchez

Academic Editor

Nigel Beebe

Section Editor

Shaden Kamhawi

co-Editor-in-Chief

Paul Brindley

co-Editor-in-Chief

**Additional Editor Comments:**

Thank you for submitting your manuscript to our journal and for your patience throughout the review process. We have now received comments from all three reviewers. One reviewer has recommended acceptance, while the other two have recommended minor revisions.

After carefully examining the full set of comments, it appears that the revisions requested—though classified as “minor” by the reviewers—are fairly substantial in scope. Several points will require clarification, additional explanation, and adjustments to the manuscript to ensure clarity, rigor, and completeness.

In light of this, the editorial decision is Major Revision Required. We invite you to submit a revised version of your manuscript that thoroughly addresses all reviewer comments. Please provide a detailed, point-by-point response outlining how each comment has been addressed. If you choose not to make a suggested change, please explain your reasoning clearly.

We value the contribution your work can make, and we believe that addressing these issues will strengthen the manuscript significantly.

Thank you again for your contribution, and we look forward to receiving your revised submission.

**Journal Requirements:**

1) Please upload all main figures as separate Figure files in .tif or .eps format. For more information about how to convert and format your figure files please see our guidelines:

2) Some material included in your submission may be copyrighted. According to PLOSu2019s copyright policy, authors who use figures or other material (e.g., graphics, clipart, maps) from another author or copyright holder must demonstrate or obtain permission to publish this material under the Creative Commons Attribution 4.0 International (CC BY 4.0) License used by PLOS journals. Please closely review the details of PLOSu2019s copyright requirements here: PLOS Licenses and Copyright. If you need to request permissions from a copyright holder, you may use PLOS's Copyright Content Permission form.

Potential Copyright Issues:

i) Figures 1, 2, 4, 5, and 7. Please (a) provide a direct link to the base layer of the map (i.e., the country or region border shape) and ensure this is also included in the figure legend; and (b) provide a link to the terms of use / license information for the base layer image or shapefile. We cannot publish proprietary or copyrighted maps (e.g. Google Maps, Mapquest) and the terms of use for your map base layer must be compatible with our CC BY 4.0 license.

**Reviewers' Comments:**

Reviewer's Responses to Questions

**Key Review Criteria Required for Acceptance?**

**Methods**

-Are the objectives of the study clearly articulated with a clear testable hypothesis stated?

-Is the study design appropriate to address the stated objectives?

-Is the population clearly described and appropriate for the hypothesis being tested?

-Is the sample size sufficient to ensure adequate power to address the hypothesis being tested?

-Were correct statistical analysis used to support conclusions?

-Are there concerns about ethical or regulatory requirements being met?

Reviewer #1: This is a good study on transmission of Dengue in Goa. The Objectives are cledar and methodologies adopted are OK.

Reviewer #2: The study objectives are clearly stated with testable hypotheses.

The study design appropriately addresses the objectives using spatial and temporal analysis methods.

The population under study is well described and relevant to the hypotheses.

Sample size sufficiency is questionable for some analyses, particularly serotype data, which has limited samples for fine-scale inference.

The statistical methods used are generally appropriate, though incorporation of underreporting adjustments and integrated data analysis could strengthen the conclusions.

There appear to be no major ethical or regulatory concerns raised based on the information provided.

Reviewer #3: - The objectives of the study are clearly articulated.

- The study design is appropriate to address the stated objectives.

- The study area is clearly described and appropriate for the hypothesis; I have suggested some improvements.

- The sample size includes data from 34 health facilities.

- The statistical analysis used supports the conclusions.

- I found no particular concerns about ethical or regulatory requirements.

**Results**

-Does the analysis presented match the analysis plan?

-Are the results clearly and completely presented?

-Are the figures (Tables, Images) of sufficient quality for clarity?

Reviewer #1: The results obtained are almost clearly analysed

Reviewer #2: The analysis mostly matches the stated analysis plan, but lacks key adjustments for underreporting and integrated data analysis as noted.

The results are generally clearly presented, but some important methodological details and model limitations should be better explained to support full interpretation.

Figures and tables are of adequate quality and clarity but could be enhanced by including quantitative values, clearer legends, and contextual explanations for greater reader comprehension.

Reviewer #3: Yes.

I have suggested improving the figures.

**Conclusions**

-Are the conclusions supported by the data presented?

-Are the limitations of analysis clearly described?

-Do the authors discuss how these data can be helpful to advance our understanding of the topic under study?

-Is public health relevance addressed?

Reviewer #1: The conclusions are supported by data

Reviewer #2: The conclusions are generally supported by the data but should be tempered by acknowledging key limitations related to data quality and methodology.

Limitations are mentioned but not fully detailed, especially regarding underreporting, serotype data constraints, and model weaknesses; these should be clearly described.

The authors discuss the potential usefulness of the data to advance understanding, but this could be expanded to include how integrated analyses might improve insights.

Public health relevance is addressed, highlighting the importance of spatial analytics for targeting interventions; however, more concrete recommendations for operationalizing findings would strengthen this aspect.

Reviewer #3: Yes

**Editorial and Data Presentation Modifications?**

Reviewer #1: The English language may be checked

Reviewer #2: Minor revision required

Reviewer #3: Yes, I have suggested minor editorial corrections in the text. See the comments file attached for detailed comments.

**Summary and General Comments**

Reviewer #1: Though small in size, Goa is one of India’s most popular states and features prominently on the wish list of countless travellers. However, it has witnessed a steep rise in dengue cases over the last decade. The authors have examined the transmission dynamics of dengue in Goa and identified DEN-2 as the dominant serotype. Most dengue clusters were concentrated in North Goa. Space–time analyses revealed a significant monotonic increase in cases within recurrent high-incidence clusters.

The objectovesof the study were addressed. And the methodologies adopted are OK.

The regression model highlighted the importance of climatic variables with a lag period of 2–3 months, as well as a rainfall threshold of 607–630 mm—rainfall above this level may lead to increased dengue transmission. The authors conclude that integrating space–time analytics, negative binomial modelling, and climate-lagged associations can produce operationally useful risk maps and short-term forecasts. These findings support pre-monsoon source reduction, targeted vector control, serotype-guided surveillance, and climate-informed early warning systems for Goa and similar settings in western India.

Reviewer #2: (No Response)

Reviewer #3: See my comments.

PLOS authors have the option to publish the peer review history of their article (what does this mean? ). If published, this will include your full peer review and any attached files.). If published, this will include your full peer review and any attached files.

**Do you want your identity to be public for this peer review?** For information about this choice, including consent withdrawal, please see our For information about this choice, including consent withdrawal, please see our Privacy Policy ..

Reviewer #1: **Yes:** Aditya Prasad DashAditya Prasad Dash

Reviewer #2: No

Reviewer #3: **Yes:** Dr Rajpal Singh YadavDr Rajpal Singh Yadav

**Figure resubmission:**
---

## [Editor Report · Decision Letter 1]

4 Mar 2026

Dear Ajeet Kumar Mohanty, PhD,

We are pleased to inform you that your manuscript 'Dengue transmission dynamics in an urban setting in western India' has been provisionally accepted for publication in PLOS Neglected Tropical Diseases.

Best regards,

Elda Eliza Sanchez

Academic Editor

Nigel Beebe

Section Editor

Shaden Kamhawi

co-Editor-in-Chief

Paul Brindley

co-Editor-in-Chief

---

## [Editor Report · Acceptance letter]

Dear Dr Mohanty,

We are delighted to inform you that your manuscript, "Dengue transmission dynamics in an urban setting in western India," has been formally accepted for publication in PLOS Neglected Tropical Diseases.

Best regards,

Shaden Kamhawi

co-Editor-in-Chief

Paul Brindley

co-Editor-in-Chief
